# Vaccination for Respiratory Infections in Patients with Heart Failure

**DOI:** 10.3390/jcm10194311

**Published:** 2021-09-22

**Authors:** Nicolas Girerd, Nicolas Chapet, Camille Roubille, Jérôme Roncalli, Muriel Salvat, Frédéric Mouquet, Nicolas Lamblin, Jean-Pierre Gueffet, Thibaud Damy, Michel Galinier, Jean-Michel Tartiere, Cécile Janssen, Emmanuelle Berthelot, Sylvain Aguilhon, Roger Escamilla, François Roubille

**Affiliations:** 1Centre d’Investigations Cliniques—INSERM CHRU de Nancy, Institut Lorrain du Cœur et des Vaisseaux Louis Mathieu, Université de Lorraine, 54500 Vandœuvre-lès-Nancy, France; n.girerd@chru-nancy.fr; 2Department of Clinical Pharmacy, CHU de Montpellier, 34295 Montpellier, France; n-chapet@chu-montpellier.fr; 3Department of Internal Medicine, CHU Montpellier, Montpellier University, PhyMedExp, 34295 Montpellier, France; c-roubille@chu-montpellier.fr; 4Service de Cardiologie, CHU de Toulouse-Rangueil, Université Paul Sabatier–Toulouse III, 31400 Toulouse, France; roncalli.j@chu-toulouse.fr; 5CHU Grenoble Alpes, 38700 La Tronche, France; msalvat@chu-grenoble.fr; 6Department of Cardiology, Hôpital Privé Le Bois, 59000 Lille, France; fmouquet@ovh.fr; 7Institut Pasteur, Université Lille, Inserm, CHU Lille, U1167 Lille, France; Nicolas.LAMBLIN@chru-lille.fr; 8Hôpital Privé du Confluent, 44000 Nantes, France; JeanPierre.Gueffet@groupeconfluent.fr; 9Referral Center for Cardiac Amyloidoisis, Department of Cardiology, GHU Henri Mondor-APHP, IMRB 955, 94000 Créteil, France; thibaud.damy@gmail.com; 10Fédération des Services de Cardiologie, CHU Toulouse-Rangueil, Faculté de Médecine, Toulouse, Université Paul Sabatier-Toulouse III, 31400 Toulouse, France; galinier.m@chu-toulouse.fr; 11CHITS, Department of Cardiology, Hôpital Sainte-Musse, 83100 Toulon, France; Jean-Michel.Tartiere@ch-toulon.fr; 12Centre Hospitalier Annecy Genevois, Infectious Diseases Unit, 74370 Annecy, France; cjanssen@ch-annecygenevois.fr; 13Service de Cardiologie, Hôpital Bicêtre, AP-HP, University of Paris Sud, 94270 Le Kremlin-Bicêtre, France; emmanuelle.berthelot@aphp.fr; 14Cardiology Department, CHU de Montpellier, 34295 Montpellier, France; s-aguilhon@chu-montpellier.fr; 15Service de Pneumologie, Hôpital Larrey, 31400 Toulouse, France; escamilla.r@chu-toulouse.fr; 16Cardiology Department, INI-CRT, CHU de Montpellier, PhyMedExp, Université de Montpellier, INSERM, CNRS, 34295 Montpellier, France

**Keywords:** pneumococcal vaccine, influenza vaccine, COVID-19 vaccine, vaccination coverage, respiratory infections, heart failure, review article

## Abstract

Bronchopulmonary infections are a major trigger of cardiac decompensation and are frequently associated with hospitalizations in patients with heart failure (HF). Adverse cardiac effects associated with respiratory infections, more specifically *Streptococcus pneumoniae* and influenza infections, are the consequence of inflammatory processes and thrombotic events. For both influenza and pneumococcal vaccinations, large multicenter randomized clinical trials are needed to evaluate their efficacy in preventing cardiovascular events, especially in HF patients. No study to date has evaluated the protective effect of the COVID-19 vaccine in patients with HF. Different guidelines recommend annual influenza vaccination for patients with established cardiovascular disease and also recommend pneumococcal vaccination in patients with HF. The Heart Failure group of the French Society of Cardiology recently strongly recommended vaccination against COVID-19 in HF patients. Nevertheless, the implementation of vaccination recommendations against respiratory infections in HF patients remains suboptimal. This suggests that a national health policy is needed to improve vaccination coverage, involving not only the general practitioner, but also other health providers, such as cardiologists, nurses, and pharmacists. This review first summarizes the pathophysiology of the interrelationships between inflammation, infection, and HF. Then, we describe the current clinical knowledge concerning the protective effect of vaccines against respiratory diseases (influenza, pneumococcal infection, and COVID-19) in patients with HF and finally we propose how vaccination coverage could be improved in these patients.

## 1. The Interrelationship between Inflammation, Infection, Thrombosis, and Heart Failure

### 1.1. Inflammation and Heart Failure

Systemic inflammation is a common feature of HF and it contributes to the progression and complications of chronic and acute HF, irrespective of left ventricular ejection fraction [1]. Thus, 57% of HF patients with preserved ejection fraction had elevated CRP in the RELAX trial [2]. Chronic inflammation participates to HF through various mechanisms, including innate and humoral immune responses [3]. The inflammatory processes lead to extracellular matrix remodeling and fibrosis responsible for the impairment of both systolic and diastolic function [1]. Iron deficiency due to chronic inflammation impairs exercise capacity and is associated with an increased risk of death independently of anemia [3]. However, the precise mechanisms and impact of chronic inflammation in HF patients remain poorly understood and no anti-inflammatory therapy, such as anti-tumor necrosis factor-α or anti-interleukin-1 therapy, has yet been shown to improve the prognosis of HF [1].

### 1.2. Respiratory Infections and Heart Failure

Inflammatory processes are activated during respiratory infections and coronary artery disease (CAD) itself appears, at least in part, as an inflammatory disease. Endothelial damage and coagulation dysregulation due to inflammation are responsible for acute events [4].

Many factors, summarized in Figure 1, potentially contribute to the adverse cardiac effects associated with respiratory infections, especially with *Streptococcus pneumoniae* and *influenza* [5]. Inflammatory processes during respiratory infections may accelerate atherogenesis through increased production of proinflammatory mediators (TNF-alpha, C-reactive proteins, or interleukins), leukocyte adhesion to vessel walls, and migration into the vascular intima. In addition, the production of cytokines during the acute inflammatory phase may alter the contractility of cardiac myocytes via a decrease in their beta-adrenergic responsiveness. Chronic and sustained production of cytokines leads to adverse remodeling of cardiac tissue. These processes have been related to left ventricular dilatation and increased collagen content, thus participating to pathophysiology of HF.

In the context of emerging respiratory infections, patients with HF are more likely to develop severe forms of coronavirus disease 2019 (COVID-19), but the pathophysiology of these mechanisms remain unclear [6]. Severe acute respiratory syndrome coronavirus 2 (SARS-CoV-2) infects respiratory cells after binding of the viral surface spike protein to the human angiotensin-converting enzyme 2 (ACE2) receptor, which is also highly expressed by cardiac cells. Besides possible direct myocardial damage through binding to the ACE2 receptor, other indirect mechanisms are suspected. Thus, the hypoxemia associated with lung injury and a storm of inflammatory cytokines (IL-6 and IL-17) due to an excessive immune response could have a toxic effect on myocardial cells leading to cardiac myocyte apoptosis [6].

### 1.3. Risk of Thrombosis in Respiratory Infections

HF is associated with an increased risk of venous thromboembolism and respiratory infections also increase the risk of thrombotic events [7]. Within one week after influenza virus infection, the risk of myocardial infarction increases 6-fold [8]. This increased risk is also observed in exacerbations of chronic obstructive pulmonary disease, which are often a consequence of infection [9].

The mechanisms underlying inflammation, infection, and thrombosis are multiple and complex and not specific of these clinical settings but could be met in other situations, such as acute coronary syndromes or thromboembolic events (Table 1) [10,11]. Inflammation enhances the procoagulant status by involving cellular and humoral factors of hemostasis (Figure 2). This results in overexpression or proinflammatory cytokines, such as IL-1 and IL-6. Proinflammatory cytokines increase tissue factor synthesis by endothelial cells and monocytes, resulting in systemic activation of coagulation. Inflammation is finally responsible for thrombin generation.

Neutrophils are a first line of innate immunity against pathogens and have recently been shown to play a role in infection-induced thrombosis through a process named NETosis [13]. During this process, there is a release of neutrophil extracellular traps (NETs) consisting of extracellular filaments, composed mainly of DNA, that bind pathogens. NETosis is activated by both pro-inflammatory cytokines and activated platelets (Figure 2). NETs have pro-coagulant properties by binding to fibrin, which facilitates trapping of platelets, red blood cells, and coagulation factors, resulting in thrombus formation. Histological studies after myocardial infarction reported that NETs were abundant in thrombus [13].

## 2. Impact of Respiratory Infections on Cardiovascular Diseases

### 2.1. Respiratory Infection with Streptococcus Pneumoniae or Influenza Virus

Bronchopulmonary infections are a major trigger of cardiac decompensation and a common cause of hospitalization in HF patients [14]. In the literature, the cause of hospitalization is usually reported as it, although various pathophysiological pathways could be intricated and although the causal link is most often likely but not certain. From a practical point of view, patients are often treated for both respiratory and cardiac disorders. Furthermore, although the population of HF patients is increasing, the prevalence and predictors of the different types of acute infections and their impact on outcome remain poorly known.

A recent sub-analysis of the PARADIGM-HF and PARAGON-HF trials reported a high incidence of pneumonia in patients with HF, especially in patients with preserved ejection fraction [15]. The incidence rates were 29 and 39 per 1000 patient-years, respectively, and were at around 3 times the expected rate. In addition, a first episode of pneumonia was associated with 4-fold higher mortality.

In the OPTIMIZE-HF study, pneumonia or another respiratory process precipitated 15.3% of cases and pneumonia was associated with higher in-hospital mortality (odds ratio, 1.60) [16]. The relationship between acute myocardial infarction and recent (within one week) influenza infection is well established [8]. However, the relationship between influenza infection and hospitalization for HF has a lower level of evidence. Thus, the severity and mortality rate of elderly patients with congestive HF increased in a retrospective cohort of 236 Japanese patients during the 1998 influenza pandemic [17]. In the retrospective SOLVD trial, patients with congestive HF had a greater risk of hospitalization during the influenza season [18]. Kytomaa et al., reported a temporal association between influenza-like activity and an increased rate of acute HF hospitalizations [19]. In a large retrospective U.S. study in 8,189,119 patients hospitalized with HF from 2013 to 2014, 0.67% had a concomitant influenza infection and were at higher risk for worse outcomes (higher in-hospital mortality, acute respiratory failure, and acute kidney injury) [20]. Taken together, these results support the evidence linking influenza infections to adverse outcomes in patients with HF.

### 2.2. Respiratory Infection with SARS-CoV-2

Patients with chronic HF are frequently elderly with multiple comorbidities and are therefore at increased risk for complications [21]. A meta-analysis in Chinese patients with COVID-19 reported that patients with hypertension and cardio-cerebrovascular diseases had a 2-fold and 3-fold increased risk of intensive care hospitalization, respectively [22]. In a study of 6439 patients hospitalized with COVID-19 in New York City, patients with chronic HF required assisted ventilation more often (3.6 times more) and had a higher risk of death (1.9 times more) [23]. In another study, a left ventricular ejection fraction <55% was associated with a worse prognosis [24]. A large retrospective study performed in the USA found that patients with acute HF hospitalized with COVID-19 were at high risk for complications and 24.2% of them died during hospitalization (compared with 2.6% for patients hospitalized with acute HF without COVID-19) [25]. In a Spanish study, patients with a history of chronic HF developed acute HF more frequently when infected with SARS-CoV-2 (11.2% vs. 2.1%) and had significantly higher mortality rates (48.7% vs. 19.0%) [26].

## 3. Current Knowledge about Vaccination and Heart Failure

### 3.1. Influenza Vaccine

Cohort studies on the efficacy of influenza vaccine in patients with HF have reported conflicting results. Indeed, most of these studies included a small number of patients, were retrospective, and assessed different outcomes. Thus, Vardeny et al., analyzed data from the PARADIGM-HF trial. Of the 8099 study participants, 1769 (21%) were vaccinated against influenza and a significant decrease in the relative risk of all-cause mortality was associated with vaccination (HR, 0.81; 95% CI, 0.67 to 0.97; *p* = 0.015) [27]. Another analysis was performed in the GWTG–HF registry, which includes hospitalized patients with a primary discharge diagnosis of HF (68% received influenza vaccination); in a subset of 64,614 patients, the one-year all-cause mortality rates in vaccinated patients were comparable to those in patients unvaccinated patients (adjusted HR 0.96, 95% CI 0.89 to 1.03, *p* = 0.25) [28]. The meta-analysis of Udell et al., analyzed 5 randomized clinical trials including 3238 influenza-vaccinated patients and 3231 unvaccinated patients [29]. Influenza-vaccinated patients had a lower risk for a composite of major cardiovascular events (2.9% vs. 4.7%; relative risk 0.64, 95% CI, 0.48–0.86, *p* = 0.003). Of interest, the effect of vaccination was greater in patients with higher-risk coronary disease (Table 2).

These results indicate the need for a large, adequately powered, multicenter trial to evaluate the efficacy of influenza vaccination in preventing cardiovascular events in patients with HF. A large randomized placebo-controlled trial (Clinicaltrials.gov NCT02762851) to test whether influenza vaccine reduces adverse vascular events compared with the control group is currently enrolling patients at high risk for cardiovascular events (NYHA functional class II–IV) [30].

Of note, the T cell response to influenza antigen is lower in HF patients than in healthy controls and, therefore, the efficacy of preventing adverse cardiovascular events may be reduced [31]. The recent INVESTED randomized clinical trial included patients (with a recent acute myocardial infarction or HF hospitalization and at least one other risk facto) who were randomized to receive either a high-dose trivalent (*n* = 2630) or standard-dose quadrivalent (*n* = 2630) inactivated influenza vaccine for up to three influenza seasons [32]. In these patients with high-risk cardiovascular disease, high-dose influenza vaccine showed no advantage over standard-dose vaccine in all-cause mortality or cardiopulmonary hospitalization (HR, 1.06; 95% CI, 0.97–1.17; *p* = 0.21).

The recent randomized double-blind placebo-controlled IAMI trial showed that influenza vaccination administered shortly after myocardial infarction reduced the risk of complications. At 12 months, the rate of the composite primary endpoint (all-cause death, myocardial infarction, or stent thrombosis) was 5.3% in the 1290 patients of the influenza vaccine group compared to 7.2% in the 1281 patients of the placebo group (HR 0.72, 95% CI 0.52–0.99, *p* = 0.040). The rates of all-cause death were 2.9% vs. 4.9% (HR 0.59, 95% CI 0.39–0.89, *p* = 0.010) and the rates of cardiovascular death were 2.7% vs. 4.5% (HR 0.59, 95% CI 0.39–0.90, *p* = 0.014) in the vaccine and placebo groups, respectively [33].

### 3.2. Pneumococcal Vaccine

A recent meta-analysis (18 studies; 716,108 participants) evaluated the protective effect of 23-valent polysaccharide pneumococcal vaccination (PPV23) in the general adult population for any cardiovascular event (myocardial infarction, HF, and cerebrovascular events) [34]. Vaccination was associated with a decreased risk of any cardiovascular event (RR 0.91; 95% CI 0.84–0.99) and myocardial infarction (RR 0.88; 95% CI 0.79–0.98) in all age groups; the effect was significant in patients aged ≥65 years but not in the <65 years age group. PPV23 vaccine was also associated with a significant reduction in the risk of all-cause mortality in all age groups (RR 0.78; 95% CI 0.68–0.88), specifically in those aged ≥65 years (RR 0.71; 95% CI 0.60–0.84). No significant reduction in the risk of cerebrovascular disease was observed after pneumococcal vaccination.

Another meta-analysis (5 studies; 163,756 participants) evaluated the protective effect of PPSV23 and/or PCV13 on all-cause mortality in adults with a history of cardiovascular disease (HF, coronary disease, cerebrovascular disease) or at a very high risk for cardiovascular disease [35]. No randomized controlled trials could be included in the meta-analysis, only observational studies. A significant decrease in all-cause mortality was found (HR 0.78, 95% CI 0.73–0.83). However, the authors emphasized the low confidence level of these results because of the risk of bias in some of the studies.

The efficacy of the pneumococcal vaccination was assessed in the GWTG–HF registry, which includes patients hospitalized with a primary discharge diagnosis of HF (66% received pneumococcal vaccination) [28]. In a subset of 64,614 patients, vaccinated and unvaccinated patients had similar rates of 1-year all-cause mortality rates (adjusted HR 0.95, 95% CI 0.89–1.01).

In a large prospective study performed in Sweden, concurrent administration of influenza and pneumococcal vaccines reduced hospitalizations for influenza and pneumonia [36]. An additive efficacy of vaccination was observed when both vaccines were administered, with a reduction in hospital admissions for influenza (37%), pneumonia (29%), and invasive pneumococcal disease (44%). An additive effect of the two vaccines was also observed for in-hospital mortality from pneumonia with a 35% reduction.

As with influenza vaccination, a large multicenter randomized clinical trial is needed to evaluate the efficacy of pneumococcal vaccination in the prevention of cardiovascular events, particularly in patients with HF.

### 3.3. COVID-19 Vaccine

No study to date, even in a subgroup analysis, has evaluated the protective effect of COVID-19 vaccine in patients with HF in terms of hospitalization, admission in intensive care units, or mortality.

## 4. Current Recommendations for Vaccination

### 4.1. Influenza Vaccination

In the United States and in China, routine annual influenza vaccination is recommended for all persons ≥6 months of age who do not have contraindications [37,38]. In Europe, vaccination recommendations vary according to countries but are generally limited to persons aged ≥65 years old or at increased risk of severe disease and influenza-related complications [39]. The European guidelines recommend annual influenza vaccinations for patients with established cardiovascular disease [40].

This has been reinforced in the recent guidelines [41], all the more as the vaccination against influenza has been well established as particularly safe. Cardiac adverse events are particularly rare. Very rare myocarditis has been reported, and the mechanisms remain controversial [42].

However, even in high-risk groups, implementation of these recommendations remains suboptimal. Of the 8099 study participants in the PARADIGM-HF trial, only 1769 (21%) received influenza vaccination with large regional variations (Netherlands, 77.5%; Great Britain, 77.2%; Belgium, 67.5%; North America, 52.8%; Asia, 2.6%) [27]. An analysis of the U.S. GWTG–HF database showed that one-third of patients hospitalized with HF did not receive influenza vaccination [28].

### 4.2. Pneumococcal Vaccination

In the United States, pneumococcal vaccination is recommended in patients older than 65 years (more precisely, recently, it has been left for the health care professional’s opinion to decide whether to vaccinate or not in the US, but not in the UK, where it remains recommended after 65 years) and earlier in high-risk immunocompetent patients, such as those with chronic cardiovascular disease (except hypertension). Based on expert opinion, the European Society of Cardiology recommends pneumococcal vaccination in patients with HF [41].

Two pneumococcal vaccines are currently available in France: the 13-valent conjugate vaccine Prevenar 13 (PCV 13), which contains 13 pneumococcal serotypes, and the 23-valent unconjugated polysaccharide vaccine Pneumo 23 (PPV 23), which contains 23 pneumococcal serotypes. The combination of the two vaccines (PCV13 and PPV23) makes it possible to obtain a higher immune response than the use of PPV23 alone.

HF is one of the clinical situations in non-immunocompromised adults predisposing to pneumococcal infection that is recognized by the French National Authority for Health (HAS) [43]. Previously unvaccinated adult individuals receive primary pneumococcal vaccination with a dose of PCV13, followed 8 weeks later by a dose of PPV23. Persons vaccinated using the previous sequence may receive a repeat injection of PPV23 after 5 years. Persons vaccinated for more than one year with PPV23 will receive PCV13 and a revaccination with PPV23 at least 5 years after the last PPV23. The necessity of subsequent revaccination is not currently established but is advised by infectious disease specialists.

The interest of vaccination for patients with HF has been reinforced in the recent guidelines [41].

### 4.3. COVID-19 Vaccination

An expert panel from the Heart Failure group of the French Society of Cardiology recently strongly recommended COVID-19 vaccination in patients with chronic HF because this population is vulnerable [44]. Indeed, HF has been clearly identified as a risk factor for poor outcome, particularly in decompensated patients or in patients with poor functional status.

Currently, at least four COVID-19 vaccines have been approved in Europe. They are expected to be safe in patients with chronic HF since no safety concerns have been raised to date in this population compared with the general population. However, to our knowledge, no specific study has been published in the HF population. The rates of serious adverse events and cardiovascular complications were comparable in the mRNA COVID-19 vaccine and placebo groups [45,46]. Nevertheless, a case series of nine patients with stage III hypertension within minutes of immunization with mRNA vaccine was reported [47]. According to the authors, more data are needed to specify hypertension rates after mRNA vaccination. In the meantime, the authors recommend blood pressure monitoring before and after vaccination in elderly patients with a history of hypertension and/or significant cardiovascular comorbidities. Although a relationship between thrombotic events and COVID-19 vaccines has been suggested, the benefit/risk ratio remains largely favorable for vaccination. Patients with chronic HF are at great risk for COVID-19 infection and these patients should be prioritized for vaccination [44].

## 5. Suggestions for Improvement of Vaccination

Data about vaccination programs or protocols in patients with HF are scarce. Classically, vaccination is considered the responsibility of primary care physicians. However, this strategy does not appear to be optimal given the low vaccination rate. We believe that, in addition to primary care physicians, cardiologists and pharmacists should also be responsible for screening and managing vaccination. In fact, every opportunity should be taken to vaccinate patients either during a routine visit or hospitalization.

Whether it is a routine office visit to renew a treatment, a cardiac decompensation, or hospitalization for another reason, such as pneumonia, the question of vaccination should be systematically raised. Any health care provider managing a patient with HF should ask if the patient’s influenza, pneumococcal, and now COVID-19 vaccines are up to date. Collaborative care programs involving both the primary care physician and the cardiologist may be even more effective [48].

One way to improve vaccination coverage is to better inform health care providers and patients. For outpatients, information can begin in the waiting room with dedicated posters or flyers, which can also be a way to initiate discussion on the subject. Patients’ adherence to vaccination recommendations also depends on the availability of the cardiologist to answer the patient’s questions and to lessen the apprehensions about vaccination. In addition, the opinion of a specialist, such as a cardiologist, may carry some weight, especially if it is consistent with the recommendations of the treating physician. Patients often trust their cardiologist and therefore may be more likely to get the vaccine if the cardiologist encourages it.

Cardiologists should be as informed as possible about recommendations for vaccination of patients, and HF and ESC guidelines for the diagnosis and treatment of acute and chronic heart failure should emphasize the usefulness of vaccination for HF patients. Lectures dedicated to vaccination should be provided during cardiologists’ initial training and continuing education sessions.

Some simple specific interventions can help increase vaccination rates, such as incorporating systematic vaccination questions into electronic medical records to identify eligible HF patients and to create automated vaccination prescriptions for nurses [28]. Annual SMS reminders on the cell phone for influenza vaccination have been reported to be effective [49]. Inclusion of vaccination in the inpatient management checklist should be considered even if the patient is hospitalized for a reason other than HF [50]. During hospitalization, nurses can play a central role in educating patients and explaining why they should be vaccinated.

The vaccination coverage of the population is very different if we consider vaccination against influenza (about 50% of patients with HF in France) and against pneumococcus (about 16% of these patients; data currently being submitted for publication). This suggests that a national health policy is required to improve vaccination coverage, involving first the general practitioner, but also other health providers, such as cardiologists, nurses, and pharmacists. Information to patients and to physicians is important, but it should be based on solid scientific evidence. Systematic recommendations addressed directly by health authorities to patients, as for influenza, could then be helpful.

## 6. Respiratory Vaccinations from the SARS-CoV2 Pandemic Perspective

Importantly, the current COVID-19 pandemic could change the game’s rules. On the one hand, prevalence and epidemiology but also the patterns of infections leading to HF decompensation could be durably impacted. On the other hand, some patients are either more skeptical towards vaccination against COVID more specifically or towards vaccination in general. In contrast, patients with chronic disease including chronic HF could be aware of the danger, as recently suggested [51].

## 7. Conclusions

Vaccination against influenza, pneumococcus, and COVID-19 are important goals in patients with heart failure, although largely underconsidered. Indeed, inflammation/infection is strongly associated with heart failure, and vaccination could offer a simple affordable protection in the frail population. However, the current vaccination rates remain low around the world and health care teams should first be more aware of this shortcoming and ways to overcome this situation, making every effort to increase these rates through multipronged approaches.

Additional large prospective studies should be led to support these efforts and bring more evidence.

## Figures and Tables

**Figure 1 jcm-10-04311-f001:**
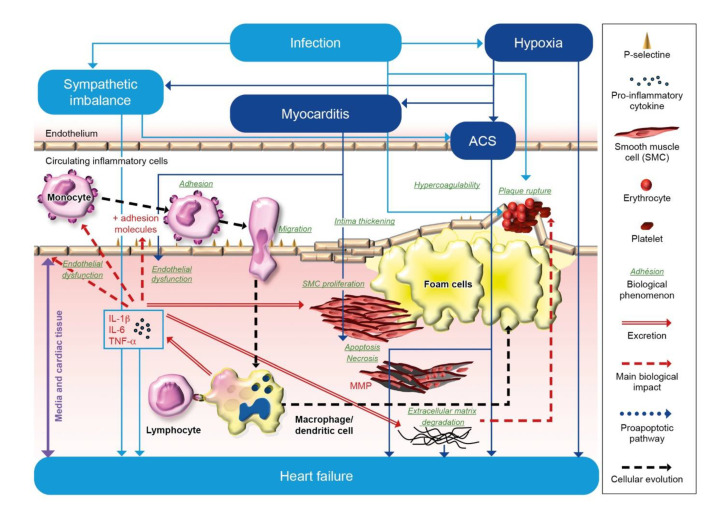
The mechanisms of the adverse effects of infection on heart failure.

**Figure 2 jcm-10-04311-f002:**
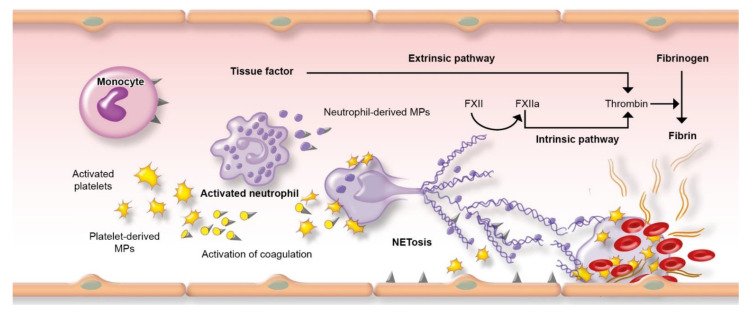
Schematic representation of thrombosis (adapted from Iba et al. [12]). Tissue factor (TF) is the pivotal initiator of the extrinsic pathway of cascade coagulation. In infection/inflammation, TF is expressed on monocytes, endothelial cells, activated platelets, and leucocytes. TF containing microparticles (MPs), mainly neutrophil- and platelet-derived MPs, represent the major source of circulating TF in plasma. NETosis occurs after stimulation of neutrophils by pathogens, activated platelets, and pro-inflammatory cytokines. NETs (neutrophil extracellular traps) are networks mainly constituted of DNA and histones that bind to pathogens and promote both the intrinsic and extrinsic coagulation pathways through the activity of serine proteases (elastase, cathepsin G) presents on NETs: serine proteases inhibit TFPI (tissue factor pathway Inhibitor), promoting TF and FXII-dependent coagulation. NETs also bind directly FXII and, in cooperation with platelets, support its activation to FXIIa leading to direct activation of the intrinsic pathway of coagulation.

**Table 1 jcm-10-04311-t001:** Mechanisms of inflammatory-induced thrombosis.

Endothelial Cell Dysfunction and Activation
Platelet activation
Modulation of plasma coagulation
Augmented pro-coagulant functions—Tissue Factor-mediated activation of coagulation
Reduction of endogenous anticoagulants: Antithrombin, Tissue Factor pathwayinhibitor (TFPI); Protein C pathway
Inhibition of fibrinolytic activity
Hyperfibrinogenemia

**Table 2 jcm-10-04311-t002:** Major adverse cardiovascular events associated with influenza vaccination vs. control in HF patients (meta-analysis of Udell et al.) [29].

	Influenza Vaccine		Placebo or Control		Risk Ratio(95% CI)
Study	Events, *n*	Participants, *n*	Events, *n*	Participants, *n*	
Govaerts et al. (1994)	7	927	5	911	1.38 (0.44–4.32)
FLUVACS trial (2004)	32	145	54	147	0.60 (0.41–0.87)
FLUCAD trial (2008)	16	325	30	333	0.55 (0.30–0.98)
De Villiers et al. (2011)	20	1620	20	1622	1.00 (0.54–1.85)
Phrommintikul et al. (2011)	20	221	42	218	0.47 (0.29–0.77)
Total	95	3238	151	3231	0.64 (0.48–0.86) ^a^

^a^*p* = 0.003 for overall effect.

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
