# Peer review of "Vaccination for Respiratory Infections in Patients with Heart Failure"

_jcm, 2021, doi:10.3390/jcm10194311_

Round 1
Reviewer 1 Report
Well written article.
The article is methodologically solid, well written without useless metanalytic virtuosity. Unfortunately in times of pandemic Sars-CoV2 loses relevance and must be contextualized.
We do not know what the real prevalence of respiratory infections is in the course of a pandemic since the use of masks and lockdowns has changed the characteristics of the classic flu pandemic.
The discussion should be reviewed in view of the recent IAMI trial, it is the largest randomized trial to date to evaluate whether influenza vaccination improves outcomes following myocardial infarction or percutaneous coronary intervention in high-risk patients with coronary artery disease.
I would add a short paragraph "respiratory vaccinations from the Sars-CoV2 pandemic perspective" and focus briefly on
- Changes in respiratory infection patterns during the pandemic
- Skepticism of some patients towards vaccination practices
- I would also mention that the flu vaccine is very safe, validated by decades of studies and experiments
- Serious side effects are rare but can happen with any drug / vaccine
- I would mention what is the risk of vaccine myocarditis after influenza vaccine
- If you can quote this article (Anatol J Cardiol. 2021; 25 (7): 522-523.
Published online 2021 Jul 1. doi: 10.5152 / AnatolJCardiol.2021.99) explaining that post-vaccination myocarditis exists but their mechanism is uncertain and often it is "serendipity".
Still great job.
Author Response
Reviewer 1
Well written article.
The article is methodologically solid, well written without useless metanalytic virtuosity. Unfortunately in times of pandemic Sars-CoV2 loses relevance and must be contextualized.
Answer. We would like to thank the reviewer for this positive evaluation.
We do not know what the real prevalence of respiratory infections is in the course of a pandemic since the use of masks and lockdowns has changed the characteristics of the classic flu pandemic.
Answer. We would like to thank the reviewer for this comment. A sentence has been added, in order to discuss this point, at the end of the discussion as follows:
“Importantly, the current COVID-19 pandemic could change the game’s rules. On the one hand, prevalence and epidemiology could be durably impacted. On the other hand, the patients could be aware of the danger, as recently suggested .”
The discussion should be reviewed in view of the recent IAMI trial, it is the largest randomized trial to date to evaluate whether influenza vaccination improves outcomes following myocardial infarction or percutaneous coronary intervention in high-risk patients with coronary artery disease.
Answer. We would like to thank the reviewer We have added a short paragraph on this recent trial (at the end of section 3.1) as follows:
The recent randomized double-blind placebo-controlled IAMI trial showed that influenza vaccination administered shortly after myocardial infarction reduced the risk of complications. At 12 months, the rate of the composite primary endpoint (all-cause death, myocardial infarction or stent thrombosis) was 5.3% in the 1290 patients of the influenza vaccine group compared to 7.2% in the 1281 patients of the placebo group (HR 0.72, 95% CI 0.52–0.99, P = 0.040). The rates of all-cause death were 2.9% vs. 4.9% (HR 0.59, 95% CI 0.39–0.89, P = 0.010) and the rates of cardiovascular death were 2.7% vs. 4.5% (HR 0.59, 95% CI 0.39–0.90, P = 0.014) in the vaccine and placebo groups, respectively.32
I would add a short paragraph "respiratory vaccinations from the Sars-CoV2 pandemic perspective" and focus briefly on
- Changes in respiratory infection patterns during the pandemic
- Skepticism of some patients towards vaccination practices
- I would also mention that the flu vaccine is very safe, validated by decades of studies and experiments
- Serious side effects are rare but can happen with any drug / vaccine
- I would mention what is the risk of vaccine myocarditis after influenza vaccine
- If you can quote this article (Anatol J Cardiol. 2021; 25 (7): 522-523.
Published online 2021 Jul 1. doi: 10.5152 / AnatolJCardiol.2021.99) explaining that post-vaccination myocarditis exists but their mechanism is uncertain and often it is "serendipity".
Still great job.
Answer.
Recent ESC guidelines are now mentioned in the main document as underlined in the new version of the manuscript.
A specific mention regarding the safety of the vaccination has been inserted as follows:
This has been reinforced in the recent guidelines , all the more as the vaccination against flue has been well established as particularly safe.
Furthermore, a new paragraph has been added at the end of the discussion, as follows:
- Respiratory vaccinations from the Sars-CoV2 pandemic perspective
Importantly, the current COVID-19 pandemic could change the game’s rules. On the one hand, prevalence and epidemiology but also the patterns of infections leading to HF decompensation could be durably impacted. On the other hand, the patients either more skeptical towards vaccination against COVID more specifically or towards vaccination in general. By contrast, patients with chronic disease including chronic HF could be aware of the danger, as recently suggested.
And in the paragraph dedicated to vaccination against influenzae
This has been reinforced in the recent guidelines, all the more as the vaccination against flue has been well established as particularly safe. Cardiological adverse events are particularly rare. Very rare myocarditis have been reported, and the mechanisms remain controversial.
- Please insert the reference PMID: 34501415
- Please insert the reference PMID: 34447992
- Please insert the reference PMID: 34501415
- Please insert the reference PMID: 34447992
Please insert the reference Anatol J Cardiol. 2021; 25 (7): 522-523.
Reviewer 2 Report
In this paper, authors reviewed the pathophysiology of the interrelationships between inflammation, infection and HF. Then, they described the clinical knowledge concerning the protective effect of vaccines against respiratory diseases (influenza, pneumococcal infection and Covid-19) in patients with HF. At the end, they proposed how vaccination coverage could be improved in these patients. The topic is really interesting and very actual.
Minor mistakes:
- please correct the first sub-title "The interrelationship between inflammation, infection, thrombosis and heart faiure"
Author Response
Reviewer 2
In this paper, authors reviewed the pathophysiology of the interrelationships between inflammation, infection and HF. Then, they described the clinical knowledge concerning the protective effect of vaccines against respiratory diseases (influenza, pneumococcal infection and Covid-19) in patients with HF. At the end, they proposed how vaccination coverage could be improved in these patients. The topic is really interesting and very actual.
Minor mistakes:
please correct the first sub-title "The interrelationship between inflammation, infection, thrombosis and heart faiure"
Answer. We thank the reviewer for his positive comments.
Sorry for the typo, the subtitle 1.1 has been corrected.
Reviewer 3 Report
Overall, this is a very good review on respiratory infections, heart failure and vaccination. The available data on this combination of subjects is minimal, and what is available may not be particularly robust. In addition, given the novelty of SARSCOV2, data is certainly limited.
Comments:
The section on thrombosis is confusing (1.3). Is the mechanism of venous thromboembolism the same as an acute coronary event/MI? The contributing inflammatory cascades certainly overlap, but one cannot assume they are the same. A clearer distinction must be made
In section 2, it is mentioned that infection “triggers” CHF, but is that true? Do we have data to suggest a causal relationship? It is certainly common to have concomitant diagnoses on presentation to the hospital and be treated for both. Often, it is difficult to distinguish the two diagnoses if there is an absence of typical PNA features. Is it also not possible, that patients develop CHF after being treated with intravenous antibiotics? Perhaps elaborate on the results from the Kytomaa et al to support this?
There is a section on thrombosis in the background pathophysiology. However, there is no data presented on the clinical incidence of venous thromboembolism in pneumonia or SARSCOV2. If there is no intention of presenting clinical data, the pathophysiology section could be eliminated for very much shortened.
In Section 5, is there evidence of successfully implemented strategies to improve vaccination in the literature, particularly in the heart failure population? If so, it may add support to statements being made.
Conclusion (section 6) must be more comprehensive/summarizing. It should highlight it as important issue, that the effects of inflammation/infection on prevalence of CHF, the underlying inherent risk of poor outcome if a patient with CHF has a bronchopulmonary reaction, indicating the need for vaccination. The current vaccination rates are low around the world and health care teams should make every effort to increase these rates through multipronged approaches.
Revisions:
Change “health actors” to “healthcare providers” throughout
In Section 3.2, the last line should be changed to “patients with heart failure” instead of “cardiovascular disease”?
Author Response
Reviewer 3
Overall, this is a very good review on respiratory infections, heart failure and vaccination. The available data on this combination of subjects is minimal, and what is available may not be particularly robust. In addition, given the novelty of SARSCOV2, data is certainly limited.
Answer. We thank the reviewer for his positive comments.
Comments:
The section on thrombosis is confusing (1.3). Is the mechanism of venous thromboembolism the same as an acute coronary event/MI? The contributing inflammatory cascades certainly overlap, but one cannot assume they are the same. A clearer distinction must be made.
Answer. We thank the reviewer. The sentence has been modified consistently as follows:
“The mechanisms underlying inflammation, infection and thrombosis are multiple and complex and not specific of these clinical settings but could be met in other situations such as acute coronary syndromes or thromboembolic events”
In section 2, it is mentioned that infection “triggers” CHF, but is that true? Do we have data to suggest a causal relationship? It is certainly common to have concomitant diagnoses on presentation to the hospital and be treated for both. Often, it is difficult to distinguish the two diagnoses if there is an absence of typical PNA features. Is it also not possible, that patients develop CHF after being treated with intravenous antibiotics? Perhaps elaborate on the results from the Kytomaa et al to support this?
Answer. We agree with the reviewer. A sentence has been added consistently as follows:
“In literature, the cause of hospitalization is usually reported as it, although various pathophysiological pathways could be intricated and although the causal link is most often likely but not certain. From a practical point-of-view, patients are often treated for both respiratory and cardiological disorders. Furthermore,…”
There is a section on thrombosis in the background pathophysiology. However, there is no data presented on the clinical incidence of venous thromboembolism in pneumonia or SARSCOV2. If there is no intention of presenting clinical data, the pathophysiology section could be eliminated for very much shortened.
Answer. We feel this section useful for the reader because the thrombosis risk has to be taken into account daily in routine practice. We would follow the opinion of the editor if he feel it useless, we would discard this paragraph.
In Section 5, is there evidence of successfully implemented strategies to improve vaccination in the literature, particularly in the heart failure population? If so, it may add support to statements being made.
Answer. We are not aware of such results. This work is the first step for this kind of approaches by our group.
Conclusion (section 6) must be more comprehensive/summarizing. It should highlight it as important issue, that the effects of inflammation/infection on prevalence of CHF, the underlying inherent risk of poor outcome if a patient with CHF has a bronchopulmonary reaction, indicating the need for vaccination. The current vaccination rates are low around the world and health care teams should make every effort to increase these rates through multipronged approaches.
Answer. The conclusion has been totally changed, following the recommendations, as follows:
“Vaccination against influenza, pneumococcus and COVID-19 are important goals in patients with heart failure, although largely underconsidered. Indeed, inflammation/infection increase the prevalence and morbimortality of HF, and vaccination could offer a simple affordable protection in the frail population. However, the current vaccination rates remain low around the world and health care teams should first be more aware of this lack and way to overcome that situation, making every effort to increase these rates through multipronged approaches.
Additional large prospective studies should be led to support these efforts and bring more evidence.”
Revisions:
Change “health actors” to “healthcare providers” throughout
Answer. done
In Section 3.2, the last line should be changed to “patients with heart failure” instead of “cardiovascular disease”?
Answer. done
Round 2
Reviewer 3 Report
Overall, thank you for your thorough responses and subsequent revisions.
It is okay to retain pathophysiology of thrombosis section as it contributes to the background mechanisms for disease.
Section 1.3: 10,11 has to be superscripted as reference
Section 2.1: Change "cardiological" to "cardiac"
Section 4.1: Change "Flue" to "influenza"
Section 4.1: Change "cardiological" to "cardiac"
Section 6: Change "game's rules" to "recommendations" or "guidelines"
Section 7: Change "increase the prevalence and morbimortality of HF" to "is strongly associated with heart failure"
Section 7: Change "lack" to "shortcoming"
Author Response
Reviewer
Overall, thank you for your thorough responses and subsequent revisions.
It is okay to retain pathophysiology of thrombosis section as it contributes to the background mechanisms for disease.
Answer. We would like to thank the reviewer for this positive evaluation.
Section 1.3: 10,11 has to be superscripted as reference
Answer. This modification has to be done by the editing service if the paper is accepted.
Section 2.1: Change "cardiological" to "cardiac"
Answer. done
Section 4.1: Change "Flue" to "influenza"
Answer. done
Section 4.1: Change "cardiological" to "cardiac"
Answer. done
Section 6: Change "game's rules" to "recommendations" or "guidelines"
Answer. We would prefer to keep this term as this means the situation and background have changed, not necessarily the guidelines.
Section 7: Change "increase the prevalence and morbimortality of HF" to "is strongly associated with heart failure"
Answer. done
Section 7: Change "lack" to "shortcoming"
Answer. done